# Protein Carbonylation As a Biomarker of Heavy Metal, Cd and Pb, Damage in *Paspalum fasciculatum* Willd. ex Flüggé

**DOI:** 10.3390/plants8110513

**Published:** 2019-11-16

**Authors:** Manuel Salas-Moreno, Neyder Contreras-Puentes, Erika Rodríguez-Cavallo, Jesús Jorrín-Novo, José Marrugo-Negrete, Darío Méndez-Cuadro

**Affiliations:** 1Biosistematic Research Group, Biology Department, Faculty of Naturals Sciences, Technological University of Chocó, Quibdó, Colombia; hasamo49@hotmail.com; 2Analytical Chemistry and Biomedicine Group, Exacts and Natural Sciences Faculty, University of Cartagena, 130015 Cartagena, Colombia; neidercontreras11@gmail.com (N.C.-P.); erodriguezc1@unicartagena.edu.co (E.R.-C.); 3Department of Biochemistry and Molecular Biology, University of Cordoba, 14014 Cordoba, Spain; bf1jonoj@uco.es; 4Universidad de Córdoba, Carrera 6 No. 77-305, Montería, Córdoba, Colombia

**Keywords:** *P. fasciculatum*, heavy metals, tolerant plant, protein carbonylation, photosynthesis proteins, mining soils

## Abstract

Heavy metal tolerant plants have phytoremediation potential for the recovery of contaminated soils, and the characterization of their metabolic adaptation processes is an important starting point to elucidate their tolerance mechanisms at molecular, biochemical and physiological levels. In this research, the effects of Cd and Pb on growth and protein carbonylation in tissues of *Paspalum fasciculatum* exposed to 30 and 50 mg·Kg^−1^ Cd and Pb respectively were determined. *P. fasciculatum* seedlings exposed to metals grew more than controls until 60 days of cultivation and limited their oxidative effects to a reduced protein group. Carbonyl indexes in leaf and root proteins reached a significant increase concerning their controls in plants exposed 30 days to Cd and 60 days to Pb. From the combined approach of Western Blot with Sodium Dodecyl Sulfate-Polyacrylamide Gel Electrophoresis (SDS-PAGE) and protein analysis by Matrix Asisted Laser Desorption/Ionisation-Time Of Flight (MALDI-TOF/TOF) mass spectrometry, chloroplastic proteins were identified into the main oxidative stress-inducible proteins to Cd and Pb, such as subunits α, γ of ATP synthetase, Chlorophyll CP26 binding protein, fructose-bisphosphate aldolase and long-chain ribulose bisphosphate carboxylase (RuBisCO LSU). Cd generated damage in the photosynthetic machinery of the leaves of *P. fasciculatum* into the first 30 days of treatment; five of the oxidized proteins are involved in photosynthesis processes. Moreover, there was a proteolytic fragmentation of the RuBisCO LSU. Results showed that intrinsic tolerance of *P. fasciculatum* to these metals reached 60 days in our conditions, along with the bioaccumulating appreciable quantities of metals in their roots.

## 1. Introduction

Cadmium (Cd) and lead (Pb) are toxic elements, which at high concentrations have various effects on the biochemistry, morphology and physiology of the plants. Lead toxicity in plants includes a wide range of metabolic alterations, damage to biomolecules, changes in membrane permeability, inhibition in activities of many enzymes, reduction in photosynthesis and transpiration and enhanced generation of reactive oxygen species (ROS) [1]. While oxidative stress induced by Cd can hinder physiological processes such as photosynthesis and respiration by inhibition of defense antioxidants systems [2,3], chlorophyll content reduction and decreases the activity of enzymes involved in CO_2_ fixation [4]. In leaves, its toxic effects include chlorosis, photosynthesis inhibition, structural and functional damages of photosystem II, disturb of mineral nutrition, intracellular redox equilibrium and water balance [2,5].

Oxidative stress promoted by Cd and Pb induces structural and functional alterations on target proteins [6]; thus, for example, induction of modification in amino-acid side chains and the main residues such as histidine, arginine and lysine, which are converted to aldehyde or keto groups [7]. The increase in the amount of these carbonyl groups per protein molecule is called protein carbonylation; this damage is irreversible because the cells are incapable to repair changes in these proteins through enzymatic reactions [7]. Carbonylation is frequently used as an oxidative stress biomarker, which is a non-enzymatic post-translational modification. Heavy metals are indirect causes of oxidative stress in plants, which is a direct cause of increased carbonylation of proteins; this has been observed in leaves exposed at Cd of species like *Arabidopsis thaliana* [8]. 

Plants have a series of strategies for heavy metal detoxification, thus reducing the adverse effects of exposure and their accumulation; this implies a complex response at molecular, biochemical, physiological and cellular levels [9]. Recent advances in the comprehension of tolerance mechanisms have been reached with proteomics, metabolomics and transcriptomics studies [10]. 

In particular, some species of genus *Paspalum* have been assayed by its potential in metals remediation such as Cd and Pb, where they have shown moderate uptake capabilities [11]. Paspalum species are distributed in tropical and subtropical regions of America, with few species in Africa and Asia. The genus presents a greater species diversity in South American countries such as Brazil, Paraguay, Uruguay, Colombia, and Argentina. In Colombia, 91 species of the genus Paspalum have been recorded [12], of which more than 20 of these species are disseminated in the Colombian Caribbean region [13], including those as *Paspalum fasciculatum* that grows in mining soils or flooded soils with a high metal load.

*P. fasciculatum* Willd. ex Flüggé is a plant with phytoremediation potential, Cd-phytostabilizer and Pb-phytoextractor that can bioaccumulate large amounts of Cd and Pb in the roots [14]; however, toxic effects of these metals on its proteome are unknown. In order to gain insights into the effects and strategies of tolerance in this plant, we used proteomics redox approaches to measure the oxidative effects of Cd and Pb exposure during the growth of *P. fasciculatum* under controlled greenhouse conditions. Thus, the carbonyl index was used as a biomarker to quantify the global oxidative damage suffered by leaves and roots proteomes of *P. fasciculatum* grown in mining soils doped with Cd and Pb along with its effects on biomass and accumulation patterns. Moreover, the carbonylated proteins in the leaves were identified as oxidative stress targets by the action of Cd and Pb in mining soils, using Western Blot analysis and tandem mass spectrometry.

## 2. Results

### 2.1. Plant Growth

*P. fasciculatum* seedlings were cultured in mining soils doped with Cd and Pb metals and growth was monitored for 90 days and expressed in terms of biomass production (Figure 1), as was previously reported [14]. It was observed that the plants in the first 60 days of growth, showed a more significant growth than the plants in control soil (*p* ≤ 0.05). By contrast, at 90 days a significant reduction in biomass in all organs (leaves, stem and roots) was recorded for plants exposed at TP50, and leaves of TC30, compared with control plants (*p* ≤ 0.05). It was found that the growth behavior of the plants examined in this study differs from that of the plants used in most studies of this type [15]. 

Plants of exposed groups suffered adverse effects on their biomass with evident symptoms of necrosis, chlorosis in leaves and a decrease in thickness in the stem between 60 to 90 days of growth; indicating that these plants have a good tolerance to exposure to Cd and Pb, at least into the first 60 days of exposure. Toxic effects observed in leaves were higher in plants exposed to Pb than in those exposed to Cd, which may be related to the high accumulation of Cd in the roots and the greatest translocation of Pb (Appendix A).

### 2.2. Cd and Pb Concentrations in Plants

As was expected, bioaccumulation of Cd and Pb in the organs of *P. fasciculatum* increased with the concentrations of these metals in the soil, following the order stems < leaves < roots (*p* ≤ 0.05) (Table 1). During the first 30 days of both exposed groups, plant organs accumulated more metal than during the subsequent 60 days of phytoremediation, with the exception of the roots in Pb50, which accumulated more Pb at 90 days. This could be interpreted as a dilution of the metals in the plant, due to significant growth concerning the control observed in some organs at 60 days.

In terms of uptake, *P. fasciculatum* showed a larger capacity to bioaccumulate Cd than Pb; its capacity to accumulate Cd in the roots is remarkable. However, this uptake is not continuous during the 90 days of exposure (except in roots of Pb50), which indicates that this species can be used to phytoremediation during short periods of time (no more than 2 or 3 months)

### 2.3. Leaves and Roots Proteins Extract Obtained

Trichlorocaetic acid(TCA)-acetone precipitation protocol [16] was used to obtain proteins from the leaves and roots of P. fasciculatum. Based on previous studies in P. fasciculatum, samples from the Cd30 and Pb50 treatments with 30, 60 and 90 days of exposure, were chosen because of the significant differences observed in biomass yielding in regard to corresponding controls. Protein extracts were used in each treatment to determine the carbonyl indexes and protein carbonylation patterns. Yielding extracts of leaves and root proteins were in the range of 0.3 to 1.0 and 0.05 to 0.6 µg/µL in TC30, respectively. For Pb50 were 0.3–1.1 µg/µL, while in controls values were between 0.60 to 1.55 and 0.08 to 0.8 µg/µL. Yielding is condensed in the Appendix A.

### 2.4. Oxidative Damage Induced by Cd and Pb in the Protein of P. fasciculatum

The determination of carbonyl indexes was carried out by building calibration curves with a standard protein of BSA and its linearity was evaluated. Figure 2 shows the dot blot membrane obtained. 

Carbonyl indexes were determined for proteins of roots and leaves of *P. fasciculatum* using the dot blots shown in Appendix A and its values are listed in Table 2. In general, it was observed an increase of protein carbonylation on roots and leaves proteomes in all samples analyzed from plants exposed to Cd and Pb (Figure 3). In particular, roots and leaves of the Cd30 treatment showed the greatest differences in oxidative damage at proteins, due to the carbonyl indexes values that increased 4.7 and 9.9 folds compared to the control, respectively. These results are congruent with the highest values of Cd absorption reached during growing experiments. For Pb50 treatment, oxidative damage in leaves was larger as the exposure time elapsed, which was coincident with the progressive biomass reduction observed. The relative increase in the carbonyl indexes for organs at the different treatments are listed in Appendix A.

### 2.5. Carbonylation Patterns of Roots and Leaves Proteins of P. fasciculatum Exposed to Cd and Pb

The qualitative pattern of carbonylated proteins was obtained for roots and leaves of *P. fasciculatum* exposed to metals, in all conditions assayed. In fact, protein carbonylated bands were not detected by the anti-DNPH antibody for the controls in experimental conditions. By contrast, carbonylated bands were detected as an effect of its exposition to Cd and Pb, which were notorious in roots and leaves exposed to Cd and leaves in contact with Pb (Figure 4), confirming the obtained by Dot blot assay. 

### 2.6. Identification of Proteins in Carbonylated Bands

To identify potentially carbonylated proteins as a consequence of Cd and Pb exposition, profiles of oxidized proteins obtained by Western blot were matched with SDS-PAGE electrophoregrams. For this, leaves protein extracts of Cd30 and Pb50 treatments exposed for 30 days and 60 days were chosen, respectively, because they showed the largest carbonyl indexes (Figure 4).

Qualitative analysis of the oxyblots obtained exhibited intense and wide protein carbonylated bands in the proteome of *P. fasciculatum* exposed to Cd and Pb. These results are consistent with the observations made in the dot blot assay. Preparative duplicates of SDS-PAGE were matched against oxyblots to identify carbonylated protein bands (Figure 5). Thus, seven protein bands were removed, which were digested with trypsin and subsequently analyzed with a MALDI-TOF-TOF spectrometer. 

Band 5, sited around 25 KDa, was common for Cd30 and Pb50, while bands 1 to 4 and 6 were only observed in TC30. A total of five proteins were identified and classified according to their function, thus: 2 ATP synthase subunit, Chlorophyll a-b binding protein, Fructose-bisphosphate aldolase, the Ribulose bisphosphate carboxylase large chain (RuBisCO LSU) and four fragments or subunits of this same protein were identified. Proteins identified are listed in Table 3 along with their accession number, mass (kDa), isoelectric point, score, submitted name, encoded on and biological process (annotated in Uniprot-Swissprot database). The proteins identified as targets of oxidative damage due to exposure to Cd and Pb are found in proteins mostly accumulated in this species due to the stress by these metals; it has been observed that these proteins are also major targets of adverse effects according to other similar research [3,17,18,19].

Furthermore, Fructose-bisphosphate aldolase and ATP synthase subunit gamma were identified in carbonylated protein bands by Cd30 and Pb50 (bands 5). Fructose-bisphosphate aldolase allosteric enzyme which plays a key role in glycolysis and gluconeogenesis. While ATP synthase subunit gamma has an essential role both in regulating the activity of ATPase and in the currents of protons across of the CF_0_ complex in the chloroplast thylakoid membrane. In the case protein bands that exhibited oxidative damage, only in treatment-Cd30 were all subunits of RuBisCO identified, which catalyzes the carboxylation of D-ribulose 1, 5-bisphosphate. This step is important for CO_2_ fixation and oxidative fragmentation of the pentose substrate in the photorespiration process. ATP synthase subunit alpha and beta, produces ATP from ADP in the presence of a proton gradient across the membrane, the alpha chain is a regulatory subunit and the beta subunits are hosted primarily in the catalytic sites. Chlorophyll a-b binding protein 1, 2, 8 and CP26, light-harvesting complex (LHC), playan important role in the reception of light, apprehension and delivery of excitation energy to photosystems, with which it has a direct relationship.

## 3. Discussion

Paspalum genus count with evidence of tolerance to heavy metals for some species such as *P. dischum* and *P. vaginatum*. For the first one, an exclusion strategy that restricts the translocation of the metals to the aerial parts has been proposed, favoring their accumulation in roots, but decreasing the availability in the soil [20]. This phytostabilizing mechanism allows it to grow well in tailings sites contaminated with Cu, Pb and Zn; hence, it has been proposed as a promising species for the repopulation of wastelands contaminated with these metals [21]. For the second one, Cd-tolerance genes have been described which could be related to the regulation of pathways involved in the synthesis of phytochelatins, heat shock factor transcription (HSFA4), protection against stress, cytochrome P450 complex (CYP450) and sugar metabolism [22].

For *P. fasciculatum,* we previously measured its capabilities to change the bioavailability of Cd and Pb in doped mining soils with these metals, its growth and development in these soils contaminated and the metals bioaccumulation in their tissues [14]. As a result, *P. fasciclulatum* is a heavy metal tolerant plant that grows in mining, flooded and dry soils, which can bioaccumulate significant amounts of Cd and Pb and increases the content of organic matter and the pH in the rhizosphere. Therefore, it can improve the quality of the soil, aiding the revegetation and colonization of other plants in such a way that it can have a significant ecological effect on the soil [14].

To deepen the physiological and biochemical aspects of the metal tolerance for this species;, in this study, we evaluated the effects of exposure to high concentrations of Cd and Pb on the development and oxidative damage on the proteome of different organs of *P. fasciculatum*. Aspects such as the growth behavior, absorption of metals in tissues and the values for carbonyl index ratios by exposure to Cd and Pb in *P. fasciculatum* (Table 2) showed differential characteristics into the time period of tolerance of the plant (30 and 60 days). The results obtained showed that the plant limited the oxidative effects to a small group of major proteins, which could be associated with the translocations of metals to the shoots where Cd and Pb induced the carbonylation of important proteins of photosynthesis, energy production, glycolysis and gluconeogenesis (Table 3).

This implies the presence of adverse effects in the energy processes, the balance of sugars, a decrease of chlorophyll and photosynthesis. Nonetheless, plants grew even more than controls up to 60 days, due maybe to the intrinsic tolerance of this plant. These plants demonstrated a marked tendency to store larger amounts of heavy metals in their roots, especially Cd. Pb presented a little more translocation to shoots, thus avoiding more severe damage to photosynthetic organs. Therefore, the mechanism of tolerance of *P. fasciculatum* would imply a decrease in the concentration of Cd and Pb in the cytoplasm.

In similar stress conditions, the synthesis of glutathione and/or phytochelatins [22,23,24], followed by compartmentalization and sequestration of metals in vacuoles, have been described along with the activation of the protective antioxidant systems [25,26].

In leaves of *P. fasciculatum*, proteins of photosynthesis were identified as the main target of the oxidative damage induced by Cd and Pb. This has been observed by other authors, who employed a diversity of abiotic stressors, including heavy metals [17,27]. Thereby, oxidative damage induced that heavy metals could force the cell to strengthen its tolerance mechanisms at the cost of growth [17], as observed in our experiment in the last 90 days. Once the tolerance period is exceeded, eventually the cells could slip into early irreversible senescence, as it was observed in the heavy metal-treated *Medicago sativa* [23], *Brassica rapa* [28] and *Nicotiana tabacum* [29]. It has been described that heavy metal effects can be reflected in the carboxylation phase of photosynthesis, mainly on enzymes of CO_2_ fixation, as RuBisCO [30]. In fact, ribulose bisphosphate of *P. fasciculatum* was identified in different protein carbonylated bands, showing similar behavior to RuBisCO of rice leaves exposed to metal stress, which increased the number of proteolytic fragments in protein electrophoresis as a consequence of the oxidative damage induced by the metals [31]. Furthermore, proteomics studies have observed that the Cd drastically reduced the amounts of RuBisCO LSU in SDS-PAGE by displacing the magnesium cofactor from its structure [19,31]. Mg^+2^ ion is the most important co-factor in carboxylation reactions catalyzed by photosynthetic enzymes as ribulose 1,5 bisphosphate carboxylase and phosphoenolpyruvate carboxylase and its removal inhibit their activity [3]. 

Likewise, in this study, other proteins affected by oxidative stress were chlorophyll (Chl) a/b binding proteins, which showed oxidative damage with Cd30 at 30 days of exposure. The light-harvesting chlorophyll a/b-binding (LHCB) proteins, are proteins with an important structural and functional role in the process of photosynthesis, it is part of the light-harvesting complex of photosystem II (PSII), which is always attached with chlorophyll A and B, and xanthophyll, thus forming the antenna complex [32]. These proteins are an important structural piece of the major light-harvesting complex, it is the membrane protein with greater distribution by a large number of plants on the planet [33]. Experimental studies show that Pb-stress is a potential inhibitor in the activity of PS II and I; being the sensitivity to Pb-stress greater in PSII than that of PSI [27]. In addition, this produces suppression of Chl biosynthesis, causing a decrease in pigment content and adverse effects on the fluorescence of Chl [27]. 

As a consequence, Cd showed greater alterations on light-harvesting chlorophyll a/b protein complex II, which causes adverse effects in the process of photosynthesis by decreasing PSII activity. This in turn causes an inhibition of quantum yield and electron transport, thus, negatively affecting the photo-assimilatory pathways, such as CO_2_ fixation, for alterations in enzymes such as RuBisCO [34]. Some studies conducted in spinach have shown differential accumulation of light-harvesting complex I and II (LHCI and LHCII) proteins in processes of adaptation to stress-by heavy metals [35]. Therefore, Cd and Pb are indirect causes of ROS, which have the capacity to produce structural and organizational damage to the lipids of the thylakoid membrane, creating an alteration in the organization of the structure of granum stacks and chloroplast ultrastructure [34,36]. Some studies indicate that PSI photoinhibition is produced by both oxygen radical (O^2−^) and singlet oxygen (1O^*2^) generated inside the thylakoid membranes; Cd and Pb causes decrease in a size and number of grana stacks, which increases the generation of ROS within the cells, directly influencing the alteration of the activity of antioxidant enzymes, and therefore altering multiproteic complexes such as PSI, PSII and ATPase in the thylakoid membrane [18,37]. Again, these toxic effects are resisted by the *P. fasciculatum* during the tolerance period.

On the other hand, it has been demonstrated that the CF_1_ CF_0_-ATP synthase (F_1_F_0_) is an important enzyme in the synthesis of ATP, starting from ADP; this protein has a crucial role in the production of energy in the cell. Structurally, it is composed of an integral membrane protein (CF_0_) and another component exposed to the stroma (CF_1_); the latter is made up of several subunits, α3β3γ1δ1ε1 [38,39]. In this research, several subunits of the ATP synthase showed oxidative damage, in Cd30 α and γ. In Pb50 only the γ subunit showed oxidative damage, which shows an effect of Cd and Pb in the alteration of energy production processes at the level of the chloroplast. Studies have shown that some ROS alter the structural components of ATP synthase, the peroxide hydrogen (H_2_O_2_) produces adverse effects on α and β-subunits; 1O^*2^ in the γ-subunit, also decreases critically the hydrolysis of ATP and the flow of protons [40,41]. 

Obtaining energy is an important physiological factor in all forms of life; therefore, glycolysis and gluconeogenesis are important biochemical processes in the development of organisms, including plants. Oxidative stress generated for both Cd30 and Pb50, oxidative damage in fructose-1,6-bisphosphate aldolase, which has a crucial role in energy metabolism, because it catalyzes rupture of β-fructose-1,6-phosphate to glyceraldehyde-3-phosphate and dihydroxyacetone in glycolysis, the process is reversed in gluconeogenesis [40]. There are reports of the adverse effects of Cd on fructose-1,6-bisphosphate aldolase enzyme activity [42]. This adverse effect of heavy metals on the important proteins of photosynthesis and energy metabolism has been observed as intolerant, hyperaccumulating and phytoremediating in plants [25,29]. 

Finally, our results show that *P. fasciculatum* has an intrinsic tolerance to the presence of Cd and Pb. In this, the intense oxidation induced by metals is limited to a group of major proteins in the first 30 days of exposure. This event perhaps activates cellular responses of the tolerance mechanism that attenuates oxidative damage and allows the plant to grow up to 60 days under the conditions of adverse cultures used.

Whereas the carbonylated proteins identified by oxidative damage in this study were those with the highest accumulation, as observed by gel electrophoresis; its carbonylation could be part of the cellular responses involved in the mechanisms of tolerance to stress by Cd and Pb in the leaves of *P. fasciculatum*.

## 4. Materials and Methods 

### 4.1. Sampling and Preparation of Soils and Growing Conditions of the Plants

The soil samples (50 Kg) were taken 30 cm from the surface at the tailings of the gold mine “El Alacrán”, located in the northwest of Colombia, between coordinates 7°44′29.01′′ North and 75°44′10.8′′ West, in the municipality of Puerto Libertador (Córdoba). Soils were stored in labeled polyethylene bags, transported to the laboratory, dried at 40 °C and sieved. Cd and Pb content was determined by atomic absorption spectroscopy (see below) and listed in Table 4, three samples were taken for each treatment. Next, three experimental groups were established. Two exposed groups, Cd30 and Pb50, corresponding to plants grown in mine soil samples doped with enough CdCl_2_ and PbSO_4_ solutions in order to prepare individual samples with 30 and 50 mg of Cd and Pb per kg of soil, respectively. Whereas a third group corresponding to plants in mine soils without doping was used as control. Cd and Pb concentrations were chosen considering the designation of unpolluted Colombian soil, which is 0.012 and 0.008 mg kg^−1^ of Cd and Pb, respectively [43]. Worldwide, these concentrations are 27 mg kg^−1^ Cd and 0.41 mg kg^−1^ Pb [44], and in countries such as the USA and Sweden, the permitted limit of heavy metal concentration in soil is 15 and 40 mg kg^−1^ Pb; 1.4 and 0.4 mg kg^−1^ Cd, respectively [45].

*P. fasciculatum* plants from soils free of Cd and Pb were cut, its taxonomic determination was carried out in the herbarium of the University of Cordoba (code HUC-8132). For the processing in the laboratory, establishment, growth conditions and monitoring of plants in doped soils and control in greenhouse conditions, we rely on what is supported by Salas-Moreno and Marrugo-Negrete [14]. To determine the amount of biomass generated (g), metal concentrations in the soil and plant organs (leaf, stem and root) were measured at three growth periods (30, 60 and 90 days).

### 4.2. Analysis of Plants and Soil Samples

Soil and plant samples were taken once a month to measure the concentrations of Cd and Pb. Organs of the plants were subjected to washing with sterile water, then they were stored in paper bags in an oven for several days at 40 °C. After that, the produced biomass was determined by weighing the exposed and control samples using an OHAUS Adventurer digital balance. For the evaluation of the symptoms of toxicity in plants such as necrosis and chlorosis, we rely on periodic observations every day, we also calculate the length and thickness of the stem every week, as well as the number of fallen leaves or with symptoms of toxicity.

For measurements of Cd and Pb concentrations in plant tissues (three samples were taken per organ of each biological replica, in each repetition), 0.5 g of plant material (dry weight) was taken, then digested with a mixture of 5 mL/2 mL of HNO_3_/H_2_O_2_ in a microwave [46]. In the same way, 0.5 g of soil (dry weight) was subjected to microwave digestion using 10 mL of 65% HNO_3_ solution according to EPA method 3051A [47]. In this process, a Milestone ETHOS TOUCH 127697 series microwave oven was carried out. The total measurements of Cd and Pb were determined by graphical oven atomic absorption spectroscopy (GF-AAS) using a Thermo Scientific iCE 3000 series analyzer. For the analytical control, certified materials were used for plants and soils (lichen: IAEA-336, 0.117 mg Cd kg^−1^ and 4.9 mg Pb kg^−1^ and soil/sediment: CRM008-050, 0.82 mg Cd kg^−1^ and 95.3 mg Pb kg^−1^).

The pipette method was used to establish the soil texture, and the texture triangle was used to establish the soil type. Cation exchange capacity (CEC) was determined by establishing the total number of removable cations (Ca^2+^ Mg^2+^ K^+^ Na^+^ Fe^3+^ Al^3+^ Mn^4+^), which were extracted with 1.0 ammonium acetate. Organic matter content (% OM) was established as the resulting sample quantity (measured from the weight of a sample), after a calcination loss of 2.0 g of soil at 450 °C for 4 h [48], while pH was determined with a WTW 330i pH meter (Table 4). All measurements were performed on three soil samples per treatment group.

### 4.3. Protein Extraction of Roots and Leaves from P. fasciculatum

Protein extraction from roots and leaves samples were obtained for control and exposed plants Cd30 and Pb50 at 30, 60 and 90 days of growth (Figure 6), in total 27 samples were analyzed by each repetition, three samples for exposure time in each organ. Organ samples were rinsed with distilled water, dried with blotting paper and cut into small pieces using a clean scissor. Samples were weighed and ground to a fine powder in liquid nitrogen using pestle and mortar [16]. 500 mg of tissue powder was macerated at 4 °C for 1 h with 2 mL of solution 10% *w*/*v* TCA in acetone supplemented with 0.07% of 2-mercaptoethanol/protease inhibitor cocktail mini-complete (Roche, BSL, Switzerland). Pellets were recovered by centrifuging (9000× *g* for 10 min at 4 °C) and washed twice with one ml of ice-cold (−20 °C) acetone containing 0.07%, 2-mercaptoethanol/protease inhibitor cocktail mini-complete. Pellets were dried at 40 °C for 10 min. Finally, proteins were extracted in 200 µL buffer lysis (5M urea/2M thiourea/1% triton-x100/50 mM Tris-HCl, pH 8.8 and 1% DTT/protease inhibitor cocktail mini-complete), vortexing for five minutes and incubated during 4 °C for 1 h. Supernatants were recovered by centrifuging at 10,000× *g* for 10 min at 4 °C. To quantify the extracted proteins, the Quick Start Bradford assay microplate (Bio-Rad, Hercules, CA, USA) was used, by generating a standard curve from BSA [49].

### 4.4. Measurement of Carbonyl Index

Irreversible oxidative damage on roots and leaves proteins of *P. fasciculatum* was quantified by Dot blot immunoassays proposed by Wehr and Levine [50], 27 samples were analyzed by repetition. To determine this, a calibration curve was built using BSA with a variety of carbonyl index values following strictly the procedure of Contreras-Puentes [51]. We prepared a solution of 2 mg/mL BSA, which was subjected to oxidation with 10 mM FeSO_4_ for 2h at 37 °C. A part of the initial BSA solution was stored to establish the basal carbonyl index value. Subsequently, the basal and oxidized solutions were derivatized with 10 mM DNPH in 0.5 M phosphoric acid, and then they were incubated for 10 min. Then, these solutions were made alkaline with 6M NaOH, incubated for 10 min and read at 450 nm [52]. The molar absorption coefficient of 22308 M-1 cm^−1^ of DNPH was determined to calculate the nmol of carbonyl/mg of protein. Stoichiometric mixing of BSA basal with BSA oxidized was performed to obtain several points on the calibration curve. Protein freshly solutions (1 mg/mL) of *P. fasciculatum* roots and leaves from treatments and control samples, were derivatized with the DNPH probe (10 mM DNPH and 2 M HCl) [53]. Derivatized BSA and samples were spotted by triplicate on PVDF membrane (Immun-Blot^®^ PVDF Membrane for Protein Blotting. Cat. #162-0177) and the membrane was immersed in a solution of Phosphate Buffered Saline (PBS) (5% skimmed milk powder dissolved in phosphate buffered saline) for 2 h at room temperature. Afterwards, they were incubated with rabbit polyclonal anti-DNPH antibody 1:10.000 (Sigma, St Louis, USA) in PBS–Tween-20 0.05% and milk 5%, for 2 h at room temperature with constant mixing movement. Then, the membranes were subjected to washing and incubation with peroxidase linked anti-rabbit IgG antibody at 1:10,000 (Amersham Bio sciences; 1h at room temperature). Chemiluminescence signals were developed using the Western maxTM rabbit IgG detection kit from Amresco and captured in a ChemiDoc System (Bio-Rad, Hercules, CA, USA). The intensity of each analyzed point was calculated through optical densitometry with Image Lab software (Bio-Rad, Hercules, CA, USA). To preserve similar measurements in the quantitative analysis, an area of 11.0 mm^2^ was determined for each protein point and the intensity measurement was organized in a matrix table in Microsoft Excel version 2013.

### 4.5. Identification of Proteins in Carbonylated Bands

Protein identification in carbonylated bands was only performed on leaves of exposed samples (Cd30 and Pb50) and control at 30 and 60-days of growth (three samples of protein bands for each treatment) because of the greatest differences in carbonyls indexes values between groups were found at these times. The use of proteomics techniques such as SDS-PAGE, Western blot and Tandem Mass Spectrometry were necessary for this investigation. In this way, DNPH-derivatized leaves proteins were electrophoresed by duplicate in 10% acrylamide/bisacrylamide gels. One of the gels was transferred to a PVDF membrane semi-dry for 30 min with a Trans-Blot turbo instrument (Bio-Rad, Hercules, CA, USA). Once the membrane was transferred, it was blocked and subjected to an incubation process with polyclonal anti-DNPH antibody, and immediately revealed by chemiluminescence, as described above [51,54]. On the other hand, the second gel was subjected to fixation in methanol 50%: 2% phosphoric acid solution and visualization of the separated proteins, which was performed by staining with Coomassie Blue Brilliant G-250. Finally, using a ChemiDoc system, immunoblot images and stained gel were visualized; in this way, the oxidized protein bands were compared and selected.

Protein bands that were chosen for the analysis were in-gel reduced, alkylated and digested with trypsin, as described in [55]. In summary, the stains were washed twice with water, reduced for 15 min with 100% acetonitrile (ACN) and dried in a Savant Speed Vac for 30 min. Afterwards, the samples were reduced with 10 mM DTT in 25 mM NH_4_ HCO_3_ (ammonium bicarbonate) for 30 min at 56 °C and then alkylated with 55 mM iodoacetamide in 25 mM NH_4_ HCO_3_ during 15 min in the dark. Samples were subjected to digestion with 12.5 ng/μL trypsin (sequencing grade, Roche Molecular Biochemicals) in 25 mM NH_4_HCO_3_ (pH 8.5) overnight to 37 °C. Finished digestion, 1 μL was spotted onto a MALDI target plate and allowed to air dry at room temperature. Then, 0.4 μL of a 3 mg/mL of the α-cyano-4-hydroxy-cinnamic acid matrix (Sigma, St Louis, USA) in 50% ACN was supplemented to the dried peptide digest spots and allowed again to air-dry at room temperature.

For MALDI-TOF/TOF, samples were analysed using an Analyzer MALDI-TOF/TOF/MS (4800 Plus Proteomics) (Applied Biosystems, MDS Sciex, Toronto, Canada), The spectrometer was operated in positive mode, the voltage acceleration used in this spectrometry was of 20 kV. All mass spectra were internally calibrated using peptides from the autodigestion of trypsin (m/z = 842.509 and 2211.104) and the peptides observed with a signal-to-noise greater than 10 were collated and represented as a list of monoisotopic molecular weights.

Through MS/MS sequencing analysis, the proteins identified by peptide mass fingerprints were processed. Taking into account the MS expectros, the most abundant precursors were chosen to undergo MS/MS analysis with collision-induced dissociation (CID) on (atmospheric gas was used) operated in 1 kV positive ion reflector mode and precursor mass windows ± 4 Da. The analysis and processing of the spectra were carried out based on the optimization of the plate model and default calibration.

The protein identification process was carried out with the help of databases, a search of each spectrum was performed in MASCOT engine v. 2.6. of the NCBI nr-database (Viridiplantae) (date 23_2018), carried out by Global Protein Server software v.3.6 from ABSciex. The parameters taken into account for the identification of proteins were the following: the modifications fixe, carbamidomethyl cysteine; the modifications variable, oxidized methionine.; peptide mass tolerance: 50 ppm for PMF, or 80−100 ppm for MS/MS or combined searches; 1 missed trypsin cleavage site; peptide charge state: +1; and MS/MS fragments tolerance: 0.3 Da. During the identification process, all proteins presented probability scores greater than the score set by MASCOT as significant (*p*-value < 0.05).

### 4.6. Statistical Analysis

The statistical analysis applied was completely random factorial, the results are presented in the experimental unit as the mean ± the standard deviation of the triplicate determinations of accumulated biomass, metals concentrations and Dot Blot assays. For data on heavy metals concentrations in the plant, organs were assessing normality using the Shapiro-Wilk test and homogeneity of variance using the Bartlett test. Later, the data were subjected to the ANOVA test, and the means comparisons were made when it was necessary using the Bonferroni tests. Statistical software GraphPad PRISM version 6.0c was used for all analyzes. A level of significance of 0.05 was selected. The significant differences between experimental groups were performed by analysis of variance in GraphPad PRISM 6.01, or between two groups by Tukey Test. Dot blot assay was triplicated.

## 5. Conclusions

*P. fasciculatum* demonstrated a tolerance capacity to stress by Cd and Pb during the process of phytoremediation in doped mining soils; this plant developed a defense mechanism reducing the concentration of metals in the cytoplasm and photosynthetic tissues and bioaccumulating almost exclusively on their roots. Good tolerance was limited in time at 60 days. The oxidative stress induced by Cd and Pb caused an increase in the carbonylation of proteins in leaves and roots of *P. fasciculatum* in treatments TC30 and TP50, at 30 and 60 days, respectively. Due to the translocation a shoots, Cd caused oxidative damage in proteins of leaves such as RuBisCO LSU, alpha and gamma units of ATP synthetase and chlorophyll binding protein CP26, exclusively affecting the photosynthetic machinery of the leaves in *P. fasciculatum* in 30 days of exposure; in addition, a proteolytic fragmentation in the subunits of the RuBisCO LSU associated with oxidative stress.

## Figures and Tables

**Figure 1 plants-08-00513-f001:**
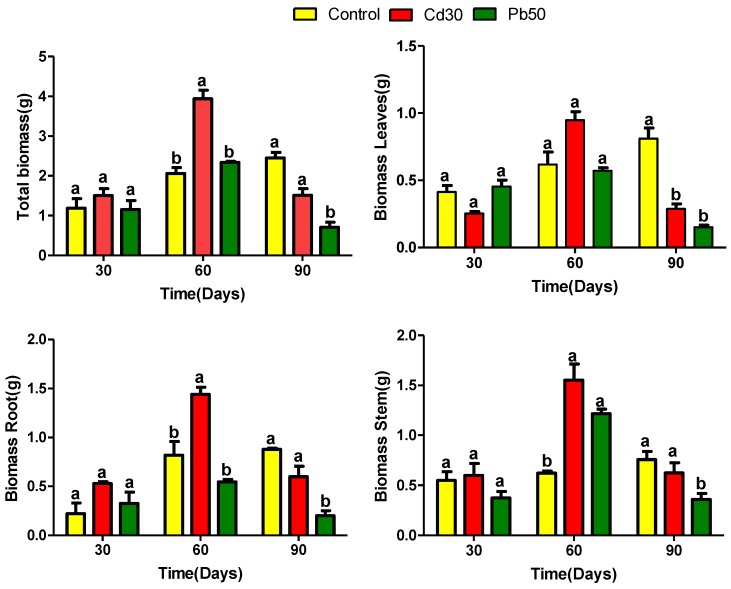
Amount of dry biomass in different tissues of *Paspalum fasciculatum* exposed to Cd and Pb in mining soils. Treatments Pb50 = 50 mg kg^−1^ Pb, Cd30 = 30 mg kg^−1^ Cd, (*n* = 3). The letters are the statistical significance (*p* < 0.05) between the control and treatments-Cd30 and Pb50.

**Figure 2 plants-08-00513-f002:**
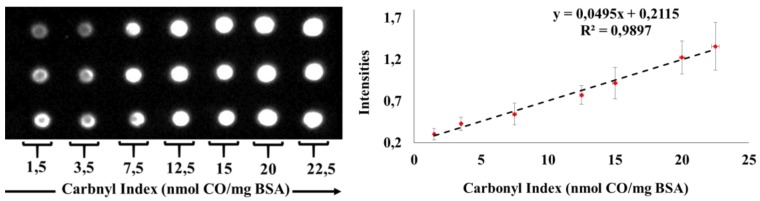
Curve of BSA for determination of carbonyl indexes employed Dot Blot assay. 200 ng of derivatized Bovine Serum Albumin (BSA) marked with 2,4-Dinitrophenylhydrazine (DNPH) were spotted by triplicate using Polyvinylidene fluoride (PVDF) membranes. Two curves were analyzed on two different days. The curves show linearity in the interval of the carbonyl index between 1.5 and 22.5 nmol of carbonyl/mg protein (R^2^ > 0.99).

**Figure 3 plants-08-00513-f003:**
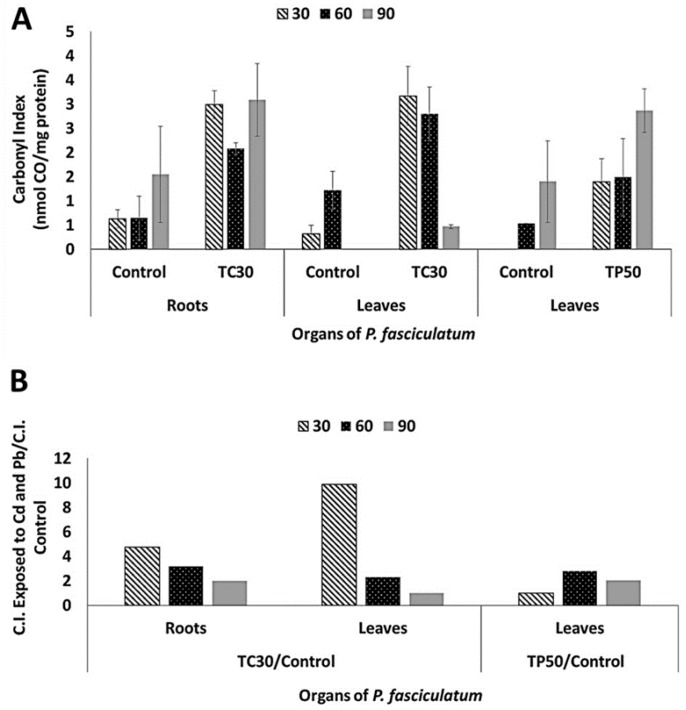
Carbonyl indexes in roots and leaves proteins of *P. fasciculatum* exposed to Cd and Pb at dissimilar days. (**a**) Evidence variation of C.I [means ± Standard Desviation (SD)] for each control, roots and leaves in exposition assays to 30 mg kg^−1^ Cd and leaves exposed to 50 mg kg^−1^ Pb. (**b**) Relation increase of C.I values for samples exposed to Cd and Pb/Control.

**Figure 4 plants-08-00513-f004:**
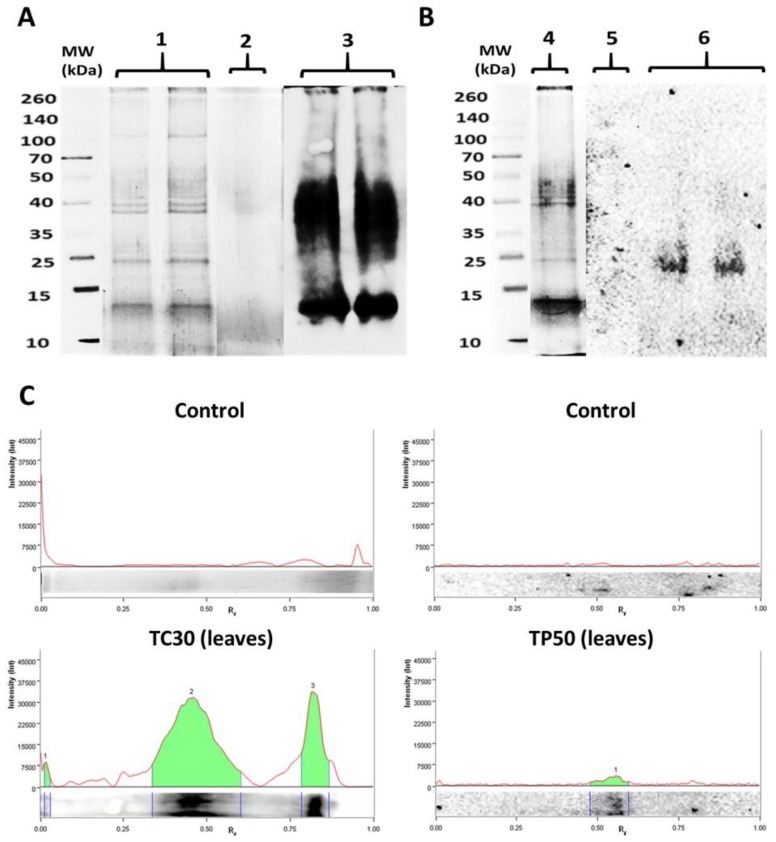
Carbonylation patterns in samples of *P. fasciculatum* exposed to Cd and Pb. (**a**) Protein profiles obtained for roots of *P. fasciculatum* exposed to 30 mg kg^−1^ Cd for 30 days: electrophoregram of Blotting of Coomassie Blue (CBB) stained proteins on SDS-PAGE (1), oxyblots for 30 days’ control (2) and exposed samples (3). (**b**) Protein profiles obtained for leaves P. fasciculatum exposed to 50 mg kg^−1^ Pb during 60 days: electrophoregram of CBB stained proteins on SDS-PAGE (4), Oxyblots for 60 days’ control (5) and exposed samples (6). (**c**) Densitograms with the intensity of the chemiluminescent signal for each carbonylated protein band obtained in oxyblots.

**Figure 5 plants-08-00513-f005:**
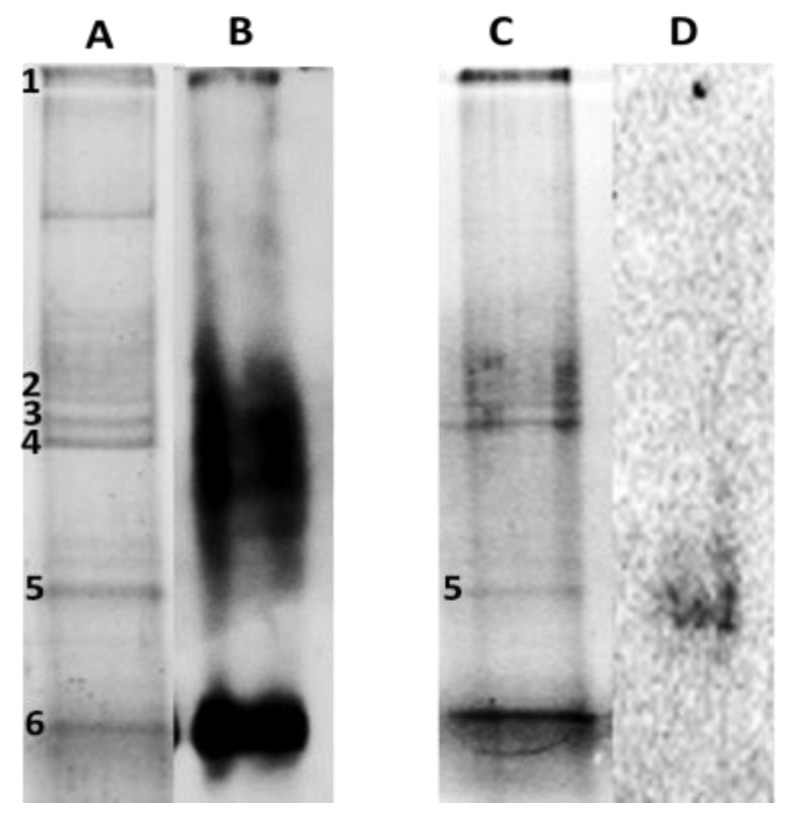
Carbonylated proteins profiles in leaves of *P. fasciculatum*. (**A**) and (**C**). Protein leaves of *P. fasciculatum* exposed Cd and Pb, electrophoresed at gel polyacrylamide staining with Coomassie brilliant blue. (**B**) and (**D**). Oxyblots of protein roots and leaves of *P. fasciculatum* exposed Cd and Pb.

**Figure 6 plants-08-00513-f006:**
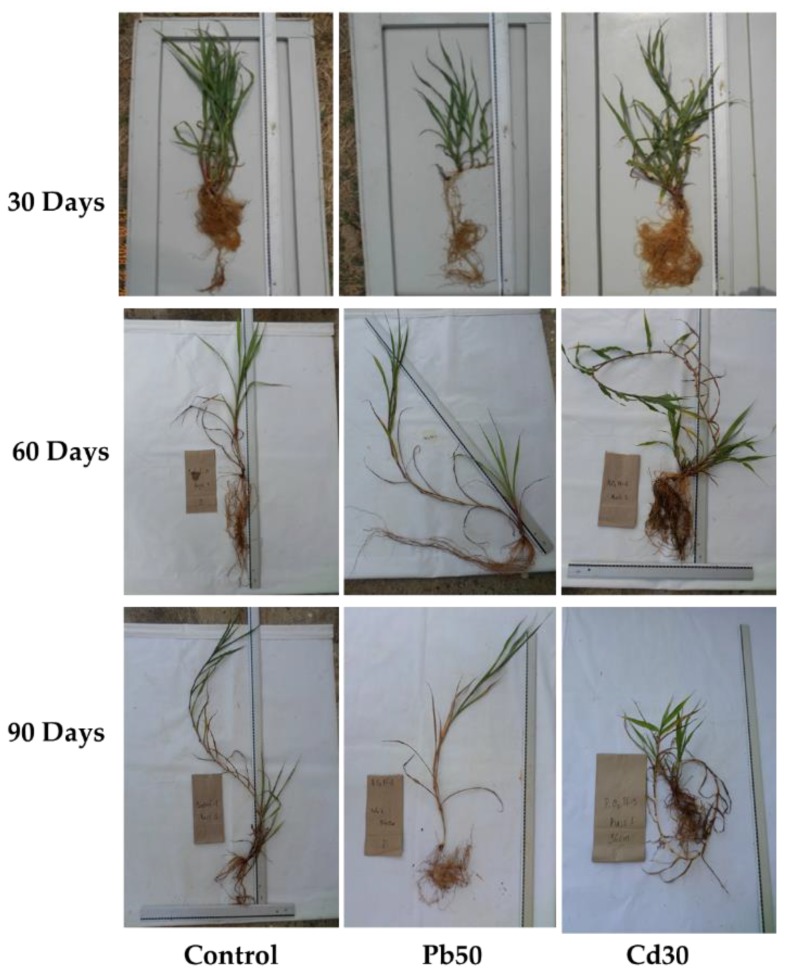
Seedlings of *P. fasciculatum* in a greenhouse under to Cd and Pb-stress for 90 days. Treatments: Cd30 = 30 mg kg^−1^; Pb50 = 50 mg kg^−1^.

**Table 1 plants-08-00513-t001:** The concentration of Cd and Pb (mg kg^−1^) in the organs of *P. fasciculatum* grown in doped mining soils.

Treatments	Plant Tissue	Cd30 (mg kg^−1^)			Pb50 (mg kg^−1^)		
		30 Days	60 Days	90 Days	30 Days	60 Days	90 Days
	Roots	190.5 ± 8 ^a^	107.1 ± 22.7 ^a^	130.8 ± 22.7 ^a^	36.7 ± 6.9 ^a^	20.8 ± 2.2 ^a^	45.7 ± 1.9 ^a^
	Stems	23.2 ± 3.8 ^a^	12.7 ± 3.67 ^a^	7.6 ± 0.7 ^a^	5.4 ± 0.6 ^a^	1.1 ± 0.02 ^a^	3.5 ± 0.1 ^a^
	Leaves	27.6 ± 5.6 ^a^	16.4 ± 7.7 ^a^	16.1 ± 0.74 ^a^	4.8 ± 2.6 ^a^	1.3 ± 0.71 ^a^	2.9 ± 0.3 ^a^
Control	Roots	2.7 ± 0.5 ^b^	3.0 ± 0.4 ^b^	2.1 ± 0.1 ^b^	0.6 ± 0.4 ^b^	1.2 ± 0.05 ^b^	0.9 ± 0.7 ^b^
	Stems	0.6 ± 0.5 ^b^	0.5 ± 0.02 ^b^	0.9 ± 0.2 ^b^	ND ^b^	ND ^a^	ND ^b^
	Leaves	1.4 ± 1.1 ^b^	1.2 ± 0.2 ^b^	1.3 ± 0.2 ^b^	ND ^b^	ND ^a^	ND ^b^

The letters are the statistical significance (*p* < 0.05) between the control and treatments-Cd30 and Pb50 in three periods of time. ND = not detectable, (*n* = 3).

**Table 2 plants-08-00513-t002:** Protein carbonyl indexes of *Paspalum fasciculatum* Willd (Ex Flugue) (Poaceae) organs.

Days	Carbonyl Index of Protein Exposed to Cd30				Carbonyl Index of Protein Exposed to Pb50	
	Roots		Leaves		Leaves	
	Control	Cd30	Control	Cd30	Control	Pb50
30	0.6 ± 0.2	2.9 ± 0.3	0.3 ± 0.2	3.2 ± 0.6	ND	1.4 ± 0.5
60	0.7 ± 0.5	2.08 ± 0.1	1.2 ± 0.4	2.8 ± 0.6	0.5 ± 0.01	1.5 ± 0.8
90	1.6 ± 0.9	3.1 ± 0.8	ND	0.5 ± 0.04	1.4 ± 0.8	2.9 ± 0.5

C.I. calculated are expressed as means ± SD (*n* = 3). Treatments Pb50 = 50 mg kg^−1^ Pb, Cd30 = 30 mg kg^−1^ Cd; ND: non-determinate. Cd: cadmium. Pb: lead.

**Table 3 plants-08-00513-t003:** Carbonylated proteins identified from *P. fasciculatum* Willd (Ex Flugue) leaves exposed at Cd (30 days) and Pb (60 days).

Band	Submitted Name		Score	Accession	Biological Process	Encoded on	Condition of Exposure
**1**	Ribulose bisphosphate carboxylase large chain [*Hordeum vulgare*]	52.0	104.0	RBL_HORVU	Catalyzes: CO_2_ fixation, oxidative fragmentation of the pentose substrate in the photorespiration process.	Plastic, Chloroplast	Cd30
**2**	ATP synthase subunit alpha [*Oryza nivara*]	55.6	157.0	ATPA_ORYNI	Translocase, ATP synthesis, Hydrogen ion transport, Ion transport	Plastic, chloroplastic	Cd30
	Ribulose bisphosphate carboxylase large chain [*Cuscuta sandwichiana*]	53.4	99.4	RBL_CUSSA	Catalyzes: carbon dioxide fixation, oxidative fragmentation of the pentose substrate in the photorespiration process	Plastic, chloroplastic	Cd30
**3**	Ribulose bisphosphate carboxylase large chain [*Avena sativa*]	52.9	150.0	RBL_AVESA	Catalyzes: carbon dioxide fixation, oxidative fragmentation of the pentose substrate in the photorespiration process	Plastic, chloroplastic	Cd30
**4**	Ribulose bisphosphate carboxylase large chain [*Avena sativa*]	52.9	150.0	RBL_AVESA	Catalyzes: carbon dioxide fixation, oxidative fragmentation of the pentose substrate in the photorespiration process	Plastic, chloroplastic	Cd30
**5**	Fructose-bisphosphate aldolase [*Oryza sativa* subsp. Japonica]	42.0	145.0	ALFP_ORYSJ	Allosteric enzyme, kinase, transferase, photosynthesis, Glycolysis; Plays a key role in glycolysis and gluconeogenesis	Cytoplasm	Cd30 and Pb50
	ATP synthase subunit gamma [*Zea mays*]	39.8	67.5	ATPG_MAIZE	ATP synthesis, Hydrogen ion transport, Ion transport, Transport, proton-transporting ATP synthase activity, rotational mechanism;	Chloroplast; chloroplast thylakoid membrane, Peripheral membrane protein	Cd30 and Pb50
	Chlorophyll a-b binding protein CP26 [*Arabidopsis thaliana*]	30.1	79.1	CB5_ARATH	light-harvesting in photosystem I, The light-harvesting complex (LHC) functions as a light receptor	Chloroplast, chloroplast thylakoid membrane	Cd30

**Table 4 plants-08-00513-t004:** Physical and chemical characteristics of the soil from “El Alacrán” gold mine.

Soil Properties					
Properties of Bioavailability		Texture		Metals	
pH	3.67 ± 0.03	Sand (%)	27.5 ± 0.03	Cd (mg kg^−1^)	7.27 ± 0.1
OM (%)	1.54 ± 0.09	Clay (%)	4.4 ± 0.09	Pb (mg kg^−1^)	2.72 ± 0.4
CEC	13.1 ± 0.01	Silt (%)	68.1 ± 0.06		

^1^ OM (%) = Organic matter percentage; CEC = cation exchange capacity.

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
