# Peer review of "Protein Carbonylation As a Biomarker of Heavy Metal, Cd and Pb, Damage in Paspalum fasciculatum Willd. ex Flüggé"

_plants, 2019, doi:10.3390/plants8110513_

Round 1

Reviewer 1 Report

The current version of the paper is well presented and structured and all the experiments have been carried out properly and the data analyzed and interpreted as expected. Considering these premises, I recommend the paper for publication after some clarify.

Comment 1: I suggest the authors change the initials of the treatments in TCd30 and TPb50 or Cd30 and Pb50 to avoid confusion with the TC (Control) or alternatively simply use the initials CTRL for the control.

Furthermore, I suggest using the same sigle for the entire manuscript (see Table 1).

Comment 2: I suggest the authors change the style of the graphs in Figure 1. I think it is easier to use only 4 graphs that simultaneously compare the variation of biomass (root, stem, leaf, total) for TC, TC30 and TP50.

Comment 3: In the results the authors assert "...decreased in the plants with a prolongation of the exposure period beyond the first 30 days" but if carefully observed the data reported in Table 1, the assert does not seem to be correct for the treatment with the Pb after 90 days in which the total quantity accumulated in the three tissues is greater than the measurement after 30 days.

Furthermore, have you carried out the analysis of the soil at the end of the experiment? These data would be appropriate to understand if it is the plant not able to accumulate high quantities of metals or if the soil has undergone changes which prevent the absorbing metals.

Comment 4: Figure 6 is mentioned in the text on line 357 after "..... TC30 and TP50 at 30, 60, 90 days of growth" but the figure shows only the plants after 30 days. Furthermore, it would be better to standardize the unit of measurement of Figure 6 with that shown in the caption of the same.

Comment 5: I suggest to the authors to insert the number of samples analyzed in each experimental methods and not only in the statistical analysis paragraph.

Author Response

We appreciate the valuable comments of the reviewer to improve our manuscript. All concerns were addressed:

Comment 1:  I suggest the authors change the initials of the treatments in TCd30 and TPb50 or Cd30 and Pb50 to avoid confusion with the TC (Control) or alternatively simply use the initials CTRL for the control.

Furthermore, I suggest using the same sigle for the entire manuscript (see Table 1).

Authors: Suggestion was accepted. Now, treatments with metals were codify as Cd30 and Pb50 for the entire manuscript. See for details:

Page 3, lines 99-100, 105, 110 Page 4, lines 114, 118,123,136,139, Tables 1, 2 and 3 Page 5, line 144 Page 6, lines 161, 175, 186; Page 7, line 190; Page 8, line 254; Page 11, lines 287-288, 295; Page 12, line 319; Page 13, lines 361, 377, Page 14, line 407.

Comment 2:. I suggest the authors change the style of the graphs in Figure 1. I think it is easier to use only 4 graphs that simultaneously compare the variation of biomass (root, stem, leaf, total) for TC, TC30 and TP50.

Authors: Figure 1 was improved. See page 3, line 93-95, figure 1.

Comment 3: In the results the authors assert "...decreased in the plants with a prolongation of the exposure period beyond the first 30 days" but if carefully observed the data reported in Table 1, the assert does not seem to be correct for the treatment with the Pb after 90 days in which the total quantity accumulated in the three tissues is greater than the measurement after 30 days.

Furthermore, have you carried out the analysis of the soil at the end of the experiment? These data would be appropriate to understand if it is the plant not able to accumulate high quantities of metals or if the soil has undergone changes which prevent the absorbing metals

Authors: Yes, indeed in the case of Pb it was different, specifically in roots; We carried out analyses of the availability of metals in the soil at the end of the process, and there were no changes that prevented the absorption of metals. Our theory proposes that the plant accumulated almost exclusively the metals in the first 30 days of exposure, then there was a decrease due to the growth of the plant, in a global context; however, the roots exposed to Pb were the meaning. Thank you for the comment, we will effectively include this exception in the phytoremediation process within the manuscript. See page 3, line 104-106.

Comment 4: Figure 6 is mentioned in the text on line 357 after "..... TC30 and TP50 at 30, 60, 90 days of growth" but the figure shows only the plants after 30 days. Furthermore, it would be better to standardize the unit of measurement of Figure 6 with that shown in the caption of the same

Authors: Yes, we very much appreciate the comment, we will make the respective changes. See page 13, figure 6.

Comment 5: I suggest to the authors to insert the number of samples analyzed in each experimental methods and not only in the statistical analysis paragraph

Authors: Information required was added. See page 12, line 317-318, line 342-343; page 13, line 357-358; line 361-362; line 381; page 14, line 407-408.

Reviewer 2 Report

The manuscript by Salas-Moreno et al. describes protein carbonylation as a biomarker under two heavy metal stresses, Cd and Pb in Paspalum fasciculatum. The methodology seems to be correct and the experiment appears to have been performed adequately by the authors. I have a few major concerns, and the authors are suggested to revise the manuscript accordingly:

Grammar and sentence structure need to be improved at many places throughout the manuscript (e.g. lines 17-19, lines 182-183, lines 303-306). In Figure 1, the authors are suggested to use different patterns of bars for TC30 and TP50 treatments. The description of the action mechanisms would allow enhancing the discussion. The authors are suggested to add photosynthetic/quantum yield data as a physiological parameter(s), if measured.

Minor issue(s) – typo errors (There are several typo errors in the manuscript and the authors are suggested to correct it during revision) – e.g. line 354: whit, line 399: wase

Replace the word ‘good’ from the conclusion section.

Author Response

We appreciate the valuable comments of the reviewer to improve our manuscript. All concerns were addressed:

Comment 1: Grammar and sentence structure need to be improved at many places throughout the manuscript (e.g. lines 17-19, lines 182-183, lines 303-306).

Authors: English grammar was revised and corrected by an expert. See page 1, line 17-20; page 6, line 181-184; page 11, line 307-310.

Comment 2: In Figure 1, the authors are suggested to use different patterns of bars for TC30 and TP50 treatments.

Authors: This figure was improved according the reviewer. See page 3, line 95-96, figure 1.

Comment 3 : The description of the action mechanisms would allow enhancing the discussion.

Authors: We will improve the description of the tolerance mechanisms. See page 8, line 231-235.

Comment 4: The authors are suggested to add photosynthetic/quantum yield data as a physiological parameter(s), if measured.

Authors: We appreciate the comments, surely it would have provided us with valuable information, but unfortunately this parameter was not measured in our experiences. This comment is very valuable, we will surely consider this parameter for our next investigations.

Comment 5: Minor issue(s) – typo errors (There are several typo errors in the manuscript and the authors are suggested to correct it during revision) – e.g. line 354: whit, line 399: wase

Replace the word ‘good’ from the conclusion section.

Authors: Thank, we appreciate the comments. See page 13, line 357, page 14, line 406; page 15, 461

Round 2

Reviewer 2 Report

The authors have considerably improved the contents of the manuscript as suggested. The manuscript can be accepted for publication in its present form.